# Deep Learning-Based Semantic Segmentation Methods for Pavement Cracks

Yu Zhang [1], Xin Gao [1,\*] and Hanzhong Zhang [2]

[1]  College of Mechanical Engineering, Shenyang University of Technology, Shenyang 110870, China
[2]  College of Professional and Continuing Education, The Hong Kong Polytechnic University, Hong Kong 999077, China
\*  Correspondence: gaoxin@smail.sut.edu.cn

**Abstract:** As road mileage continues to expand, the number of disasters caused by expanding pavement cracks is increasing. Two main methods, image processing and deep learning, are used to detect these cracks to improve the efficiency and quality of pavement crack segmentation. The classical segmentation network, UNet, has a poor ability to extract target edge information and small target segmentation, and is susceptible to the influence of distracting objects in the environment, thus failing to better segment the tiny cracks on the pavement. To resolve this problem, we propose a U-shaped network, ALP-UNet, which adds an attention module to each encoding layer. In the decoding phase, we incorporated the Laplacian pyramid to make the feature map contain more boundary information. We also propose adding a PAN auxiliary head to provide an additional loss for the backbone to improve the overall network segmentation effect. The experimental results show that the proposed method can effectively reduce the interference of other factors on the pavement and effectively improve the mIou and mPA values compared to the previous methods.

**Keywords:** attention module; Laplacian pyramid; PAN

## 1. Introduction

Pavement crack is a common pavement disease, and it is often not brought to our attention and is generally only considered by us as a normal phenomenon of pavement aging [1]. However, traffic accidents, such as pavement collapse due to the continuous expansion of pavement cracks, occur frequently. With the erosion of rainwater and the crushing of vehicles causing the increase of pavement cracks, if these cracks are not repaired in time, they will affect traffic safety, and in more serious cases even lead to landslides on panhandle roads. If road maintenance intervenes in the early stage of pavement damage, not only can it reduce the repair cost and repair time, but it can also greatly extend the service life of the road and reduce the disaster caused by the aging of the road. The traditional manual maintenance method is risky and inefficient. It requires a lot of human and material resources, which makes it difficult to complete many road maintenance tasks in a timely manner [2]. In contrast, an automated pavement crack detection system developed using computer vision and deep learning technologies can do the job quickly and accurately while eliminating the subjective factor [3]. Such systems can use low-cost devices such as smartphones or drones to capture high-resolution images and identify crack locations and types through methods such as deep convolutional neural networks and adaptive threshold segmentation. This will not only improve the efficiency and accuracy of detection but also reduce labor costs and risks. Therefore, road maintenance should be developed in the direction of intelligence and efficiency.

In the field of computer vision-based pavement crack detection and segmentation [4], the research direction is broadly divided into two parts. One is based on image processing, which mainly focuses on manual recognition of the collected data [5], using a variety of feature rules such as HOG (Histogram of Oriented Gradient), frequency, greyscale, edge,

texture, entropy, etc. and then designing some feature recognition conditions for recognition. The second is to establish a convolutional network based on deep learning to extract features from the dataset [6] and make the network continuously self adjust according to a specific loss function to achieve output data equal or approximate to the label.

We next provide a comparative review of previous studies on pavement crack segmentation, which can be divided into two main categories, i.e., image processing-based and deep learning-based approaches.

### 1.1. Image Processing-Based Methods

In the early research, crack detection methods mainly combined or improved traditional image processing techniques, such as threshold processing and edge detection. Furthermore, an automatic detection method for pavement cracks was generated. However, the result obtained by this method is the centerline of the crack, which does not include the width of the crack. In order to get more information about cracks, image processing technology is used to preprocess images, segment images, and extract features, and a fast, automatic detection and segmentation method is developed. Shi, Y. et al. [7] proposed a new forest structure-based road crack detection system, CrackForest, to address the problems of severe crack inhomogeneity, complex topology, texture similarity, and noise, and the experimental results proved that CrackForest has advanced detection accuracy. Oliveira and Correia [8] performed appropriate smoothing of the data images to reduce false positive detection results, and then iteratively classified the binary pixels into cracked and noncracked classes to identify intact pavement cracks. Lu et al. [9] proposed a pavement crack identification method based on automatic threshold iteration. They improved the peak threshold selection method by completing image enhancement, smoothing, and denoising processes before iterative threshold selection. The improved peak threshold selection method could realize real-time automatic threshold selection and ensure the stability of the detection process. Dinh, T.H. and Ha, and Q.P. et al. addressed the potential problem of concrete crack detection by using a threshold histogram approach to extract regions of interest from the background [10].

### 1.2. Deep Learning-Based Methods

With the technological breakthrough of deep learning in recent years, detection algorithms based on deep learning and convolutional neural networks have achieved better results in pavement crack identification.

Zhang, L. first applied deep learning to pavement crack detection [11]. In order to classify cracked images, he demonstrated that using convolutional neural networks is superior to supporting vector machines in classifying cracked images. However, the designed network is primitive, the training data of the image scenes are not complete enough, and the recognition efficiency and accuracy are low. Henrique Oliveira et al. [3] proposed a fully integrated system for the automatic detection of pavement cracks, which eliminates the need for manual labeling of samples, minimizes the human subjectivity generated by traditional visual inspection, and achieves crack detection and crack-type classification based on image blocks. Cha, Y.J. proposed a crack detection algorithm incorporating sliding windows and convolutional neural networks [6], which reduces the disturbance of image cutting on crack recognition. Experimental results show that the algorithm outperforms Sobel edge detection and canny-edge detection and can achieve higher crack classification accuracy. Zhang, A. and Wang, K.C.P. et al. [12] proposed an efficient architecture for CrackNet based on convolutional neural networks (CNN). It uses CNN to predict the class of each pixel of an image, which is significantly better than the state-of-the-art image processing methods and machine learning-based classifiers, though there is a need for further improvement to detect finer cracks. VishalMandal proposed an automatic pavement disease analysis system based on YOLOv2 [13]. The system obtained the final average score through the correct rate and recall rate to evaluate the classification and detection accuracy of the proposed distress analyzer. König, J. proposed

an architecture based on a fully convolutional network, UNet, and a residual module [14] and achieved good segmentation results for crack images. Garbowski and Gajewski [15] proposed a semiautomatic inspection tool based on 3D profile scan data that can identify and measure defects such as cracks, potholes, and crumbling on road surfaces. They used techniques such as laser scanners, image processing techniques, support vector machine classifiers, and a human–computer interface. A fast, accurate, and visual assessment of the road surface condition was achieved and compared with manual inspection methods. Its effectiveness and accuracy were demonstrated. Seo, H. and Huang, C. improved the U-Net by proposing mU-Net [16], introducing a feature extraction module between the encoder and decoder, fusing high-level features with low-level features, and using a region-based growth method to generate an initial liver segmentation mask to improve segmentation accuracy and robustness. Combining gradient descent and Newton's method accelerates network convergence and avoids local optimal solutions.

These inspection methods use different schemes to process and identify cracked images. Each detection system has a specific scope of application. Most of the existing detection techniques are designed to accomplish the classification of crack types and pavement damage assessment. However, due to the good crack resistance and stability of asphalt pavements, cracks are mostly small and located on rough, gray, and disturbed surfaces, this affects the accuracy of the algorithm to varying degrees. In practical applications, the existing methods still have the following shortcomings. For example, it is not effective in segmenting small cracks and is prone to misjudgment of road surface disturbances such as manhole covers and oil stains. For the edge of the lane line, it is easy to produce an oversegmentation problem. In this paper, these characteristics of pavement crack images are analyzed and a novel method based on deep learning networks is proposed. The method can simultaneously improve the segmentation effect in terms of crack details and reduce the effect of interfering objects on the segmentation effect. Extensive experiments are also conducted on a test dataset with representative features, and significant advantages are achieved in several metrics compared to existing methods. This paper provides a novel and effective technical means for the field of road surface crack image analysis, and also provides insight and reference for other similar problems.

## 2. Proposed Method

This section describes the structural components of the algorithm after the improvement and explains the reasons for the modification. The focus of this paper is to improve the accuracy of the model to segment the crack data in the presence of disturbances. With the Unet model [17] as the main backbone and the CBAM (convolutional block attention module) attention module [18] added to each layer of the encoder (left path) to better perform initial feature extraction on the input data, the decoder structure concatenates Laplacian residuals [19] based on Unet to incorporate more detailed information.

### 2.1. CBAM-Unet Model

Different from the SE (squeeze-and-excitation) attention module [20], the CBAM module infers attention feature maps along two separate dimensions, channel and spatial attention mechanisms [18]. Channel attention determines "what is important", and spatial attention determines "where is important". It provides adaptive feature refinement and improves the representation of interests. This is helpful for small objects and difficult samples. Compared with BAM (bottleneck attention module) [21], it is used in the bottleneck and can be used as a plug-and-play module for any intermediate convolutional layer module.

From the perspective of data features, pavement cracks belong to small objects, and the area ratio between the foreground and the background is very unbalanced. In order to enable Unet to better extract features from details in data in limited data, we proposed CBAM-Unet which adds an attention module before the down-sampling of the encoding layer of the contraction path. In this way, the feature map obtained by convolution is weighted in the channel dimension and spatial dimension, which enables the network

to pay more attention to the target region when extracting features. This process can be formulated as follows:

$$\begin{cases} E_i = \text{DouConv}(x), \\ E_{i+1} = \text{Att}(E_I), \end{cases} \tag{1}$$

where $E_i$ denotes the i-th encoding layer of the network model, and x is the feature map input from the previous layer. DouConv($\cdot$) denotes repeated application of two $3 \times 3$ convolutions (unpadded convolutions), and Att($\cdot$) is the CBAM attention module. The decoding layer includes a deconvolution layer and a convolutional layer to restore the feature map after convolutional downsampling to its original detail and size. On this basis, we add weight standardization to the preactivation convolution blocks [22] of the decoder architecture, which helps slightly improve the mIou indicator.

### 2.2. Laplacian Pyramid

The Laplace pyramid has been applied in various fields of scene understanding since it preserves local information about given data [23] and it emphasizes the differences between different scale spaces. This feature is exactly what is missing in the higher-order features generated in the encoding phase. After applying the Laplace pyramid transform to the image, the results at different scales contain more distinct boundary information [24]. Incorporating this boundary information into the decoding stage can improve the segmentation of small objects by the network structure. The decoding process incorporates multiscale Laplace features of the input three-channel color image. The structure of the proposed method is shown in Figure 1. Based on this decoding scheme containing the multiscale Laplacian residuals of the input images, this coding structure can be more effective for feature extraction of fine cracks.

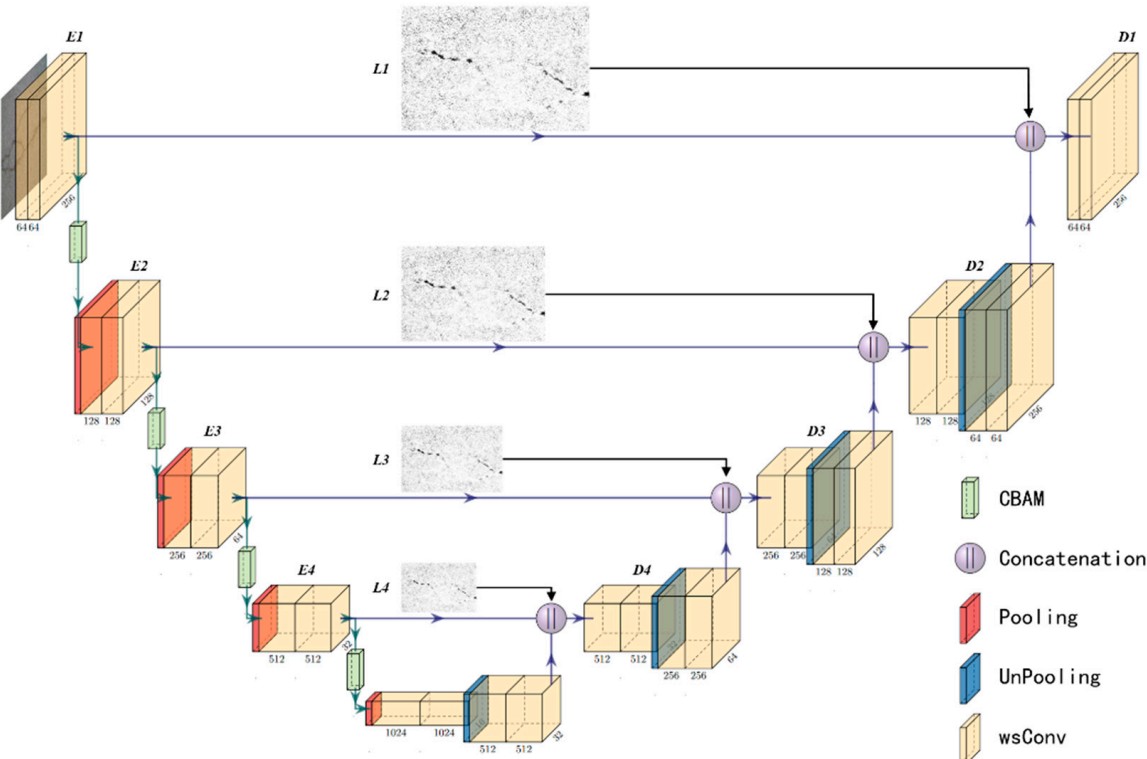

**Figure 1.** AL-Unet combining CBAM and Laplace pyramid.

Calculate the Laplacian residual on the input image:

$$L_k = I_k - \text{Up}(I_{k+1}), \ k = 1, 2, 3, 4 \tag{2}$$

The k in the above equation denotes the number of layers in Laplacian pyramid, $I_k$ is obtained by downsampling the original input image by $1/2^{k-1}$, and $Up(\cdot)$ represents the image resize using bilinear interpolation.

$$D_k = cat(E_k, L_k, UpwsConv(D_{k+1})), \ k = 1, 2, 3, 4 \tag{3}$$

After encoding and decoding calculation, $D_k$ has the feature information of different levels of encoder and the boundary features of Laplacian residuals. $UpwsConv(\cdot)$ denotes preactivation convolution blocks with weight standardization. $D_1$ is calculated as the final feature output of the lead head for loss calculation. We called the UNet combining CBAM and Laplace pyramid AL-UNet.

*2.3. PAN Path-Aggregation Auxiliary Head*

Deep supervision is a commonly used technique in deep network training. Its main idea is to add additional auxiliary heads in the middle layer of the network, which is used as an auxiliary loss to guide the weight of the shallow network. In this paper, AL-UNet is used as the main feature extraction module, and PAN [25] is used as the auxiliary head. We modified the input and output of the original PAN without changing the performance, and D2, D3, and D4 in the decoder stage of AL-UNet are used as the input of the PAN feature extraction module to extract auxiliary features from the input image. Di(i = 1, 2, 3, 4) also has shallow, deep features and boundary information of the Laplacian. The three outputs of PAN were spliced at the same scale and passed through a layer of two $1 * 1$ CBL to provide auxiliary loss for AL-UNet. The architecture of the PAN is shown in Figure 2.

*2.4. Loss Function*

The pavement crack segmentation task is a typical fine-crack segmentation task, and its data is characterized by an extremely unbalanced proportion of positive and negative samples, i.e., a very low proportion of crack categories and a high proportion of background categories. In this case, if only BCELoss [26] is used as the loss function, it may cause the network to overfit the background category and ignore the detection of the crack category. Therefore, other loss functions need to be used to supplement BCELoss to improve the network's focus on the crack category and segmentation accuracy.

DiceLoss [27] is a loss function based on the proportion of overlapping regions, which measures the similarity between segmentation results and labels, and is balanced for positive and negative samples. In the pavement crack segmentation task, using only BCELoss as the loss function may cause the network to be insensitive to the differences between positive and negative samples due to the extremely unbalanced ratio of positive and negative samples. Using DiceLoss allows the network to pay more attention to the differences between positive and negative samples and optimize the proportion of overlapping regions between segmentation results and labels.

FocalLoss [28] is an improved cross-entropy loss function that dynamically adjusts the weights according to the difficulty of the samples, allowing the network to focus more on hard-to-classify samples and less on easy-to-classify samples. In the pavement crack segmentation task, the crack category tends to be the hard-to-classify samples because of its low proportion, while the background category tends to be the easy-to-classify sample due to its high proportion. Using FocalLoss can make the network focus more on the detection of crack categories, thus improving the recall and F1 value.

Therefore, in the pavement crack segmentation task, the loss function is chosen according to the characteristics of the difference between foreground and background scales as follows:

$$BCELoss(y_i, p_i) = -\frac{1}{N} \sum_{i=1}^{N} [y_i \log(p_i) + (1 - y_i) \log(1 - p_i)] \tag{4}$$

$$DiceLoss(y_i, p_i) = 1 - \frac{2 \sum_{i=1}^{N} y_i p_i + \varepsilon}{\sum_{i=1}^{N} y_i + \sum_{i=1}^{N} p_i + \varepsilon} \tag{5}$$

$$\text{FocalLoss}(p_i) = -\frac{1}{N}\sum_{i=1}^{N}(1-p_i)^{\gamma}\log(p_i) \tag{6}$$

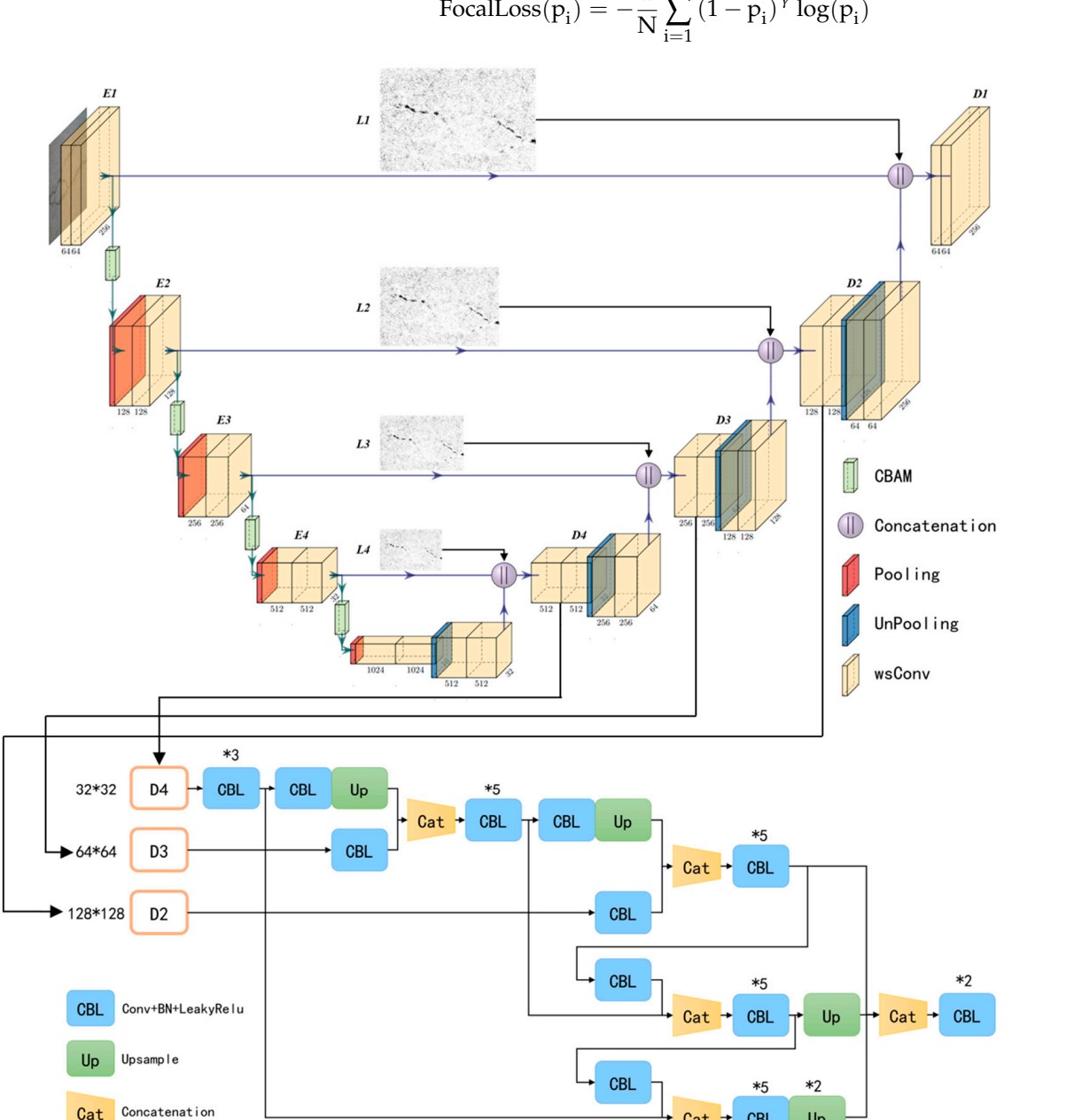

**Figure 2.** The ALP-UNet structure.

The $y_i$ in the above equation is the true value of the ith pixel in the pavement crack image, $p_i$ is the probability value of the network prediction of the ith pixel prediction, and N is the number of pixels in the image data. $\varepsilon$ in DicelLoss is a smoothing term to prevent the denominator from going to zero. The $\gamma$ in FocalLoss is the weight parameter that adjusts the hard and easy samples. When the sample is an easy-to-classify sample, the $p_i$ is large, and the loss of easy-to-classify samples is significantly reduced after adding the $\gamma$ power. The $p_i$ of the hard-to-classify sample is around 0.5, and obviously, its loss is reduced much less than that of the easy-to-classify sample.

In this segmentation task, we set the weights of BCELoss, DiceLoss, and FocalLoss as shown in Equation (7). The reason for setting equal weights for BCELoss and DiceLoss is that the objectives of the two loss functions are different, BCELoss is more concerned with the classification accuracy, while DiceLoss is more concerned with the similarity of the predicted segmentation results to the true segmentation, so the equal weights of these

two loss functions can balance these two objectives. FocalLoss can enhance the performance of the model by improving the learning of hard-to-classify samples. The reason for setting the weight of FocalLoss to two is that there are a large number of hard-to-classify samples in the pavement crack dataset of this paper, so the model needs to be enhanced to learn these samples to improve the segmentation accuracy.

$$L = L_{BCE} + L_{dice} + 2L_{focal} \tag{7}$$

## 3. Results

In this section, 150 road surfaces containing manhole covers, oil stains, and lane lines were selected for experiments with different network structures to evaluate the performance of the proposed method.

### 3.1. Training

The proposed method was tested on the ubuntu18.04 system with AMD Ryzen 7 5800 H with Radeon Graphics CPU @3.20 GHz and an NVIDIA GeForce RTX 3070 Laptop GPU. It is implemented on the PyTorch [29] framework. The network uses the SGD optimizer [30]. This network was trained from scratch for 100 epochs using an SGD optimizer with a batch size of four, where the learning rate is 0.01, momentum is 0.9, and the weight decaying factor is 0.0005. The polynomial decay was set to power 0.9 and the minimum learning rate to $10^{-4}$.

In the training phase, we used 700 cracked asphalt pavement images of $480 \times 320$ pixels size that we acquired ourselves as training data. The data was mostly fine cracks and contained distractions such as manhole covers, oil stains, and lane lines. Online data enhancement was performed during training to reduce the effect of overfitting problems. The training samples were randomly resized with a multiplier of 0.5 or 2 and randomly trimmed from $480 \times 320$ pixels to $256 \times 256$ pixels. The input image also flipped horizontally with a probability of 0.5. Moreover, photometric distortion was also added, such as adjusting the brightness, chroma, contrast, and saturation of the image and adding noise.

### 3.2. Performance Evaluation

The model uses UNet as the backbone network for feature extraction. Based on the characteristics of the asphalt pavement dataset used, UNet was selected to be combined with CBAM and a Laplacian Pyramid and trained using PAN as an auxiliary head.

The task of pavement crack segmentation was pixel-level classification to judge whether a pixel is a crack or a pavement, crack is the target that needs to be detected and segmented, which are called positive classes, and others are called negative classes. By comparing the segmentation result with the real value, true positives $N_{TP}$, false positives $N_{FP}$, true negatives $N_{TN}$, and false negatives $N_{FN}$ in the confusion matrix can be obtained. $N_{TP}$ is the number of pixels that correctly classify a crack class into a crack class. $N_{FP}$ is the number of pixels that misclassify noncrack class as a crack class. $N_{TN}$ is the number of pixels that classify the noncrack class as a noncrack class. $N_{FN}$ is the number of pixels that misclassify cracks into noncracks. In order to evaluate the results of pavement crack segmentation, the check-all rate recall, the check-accuracy rate precision, the summation average mFscore, the average pixel accuracy mPA, mean intersection over union mIou, and the dice coefficient are selected as evaluation indexes. Their calculation formulae are as follows:

$$Recall = \frac{TP}{TP + FN} \tag{8}$$

$$Precision = \frac{TP}{TP + FP} \tag{9}$$

$$mFscore = \frac{1}{k + 1} \sum_{i=0}^{k} \frac{2 \times Precision_k \times Recall_k}{Precision_k + Recall_k} \tag{10}$$

$$\mathrm{mPA} = \frac{1}{k+1} \sum_{i=0}^{k} \mathrm{Precision}_k \qquad (11)$$

$$\mathrm{mIou} = \frac{1}{k+1} \sum_{i=0}^{k} \frac{TP}{TP + FP + FN} \qquad (12)$$

$$\mathrm{Dice} = \frac{2TP}{2TP + FP + FN} \qquad (13)$$

We selected several open-source segmentation algorithms UNet, UperNet, ResUNet, and Pointrend with good segmentation effects and compared the segmentation effects on the collected pavement segmentation dataset. We used the same crop size and batch size and performed the same number of iterations of training, and the segmentation results are shown in Table 1 and Figure 3. From the results, we can see that Pointrend has a certain effectiveness.

**Table 1.** Quantitative evaluations on the pavement crack dataset.

| Architectures | Recall | Precision | mFscore | mPA | mIou | Dice |
|---|---|---|---|---|---|---|
| UNet | 0.7595 | 0.5273 | 0.8089 | 0.7624 | 0.7213 | 0.6225 |
| UperNet | 0.7369 | 0.5543 | 0.8144 | 0.7731 | 0.7261 | 0.6301 |
| ResUNet | **0.8261** | 0.5265 | 0.8192 | 0.7625 | 0.7324 | 0.6431 |
| Pointrend | 0.7559 | **0.5717** | **0.8234** | **0.7846** | **0.7372** | **0.6510** |

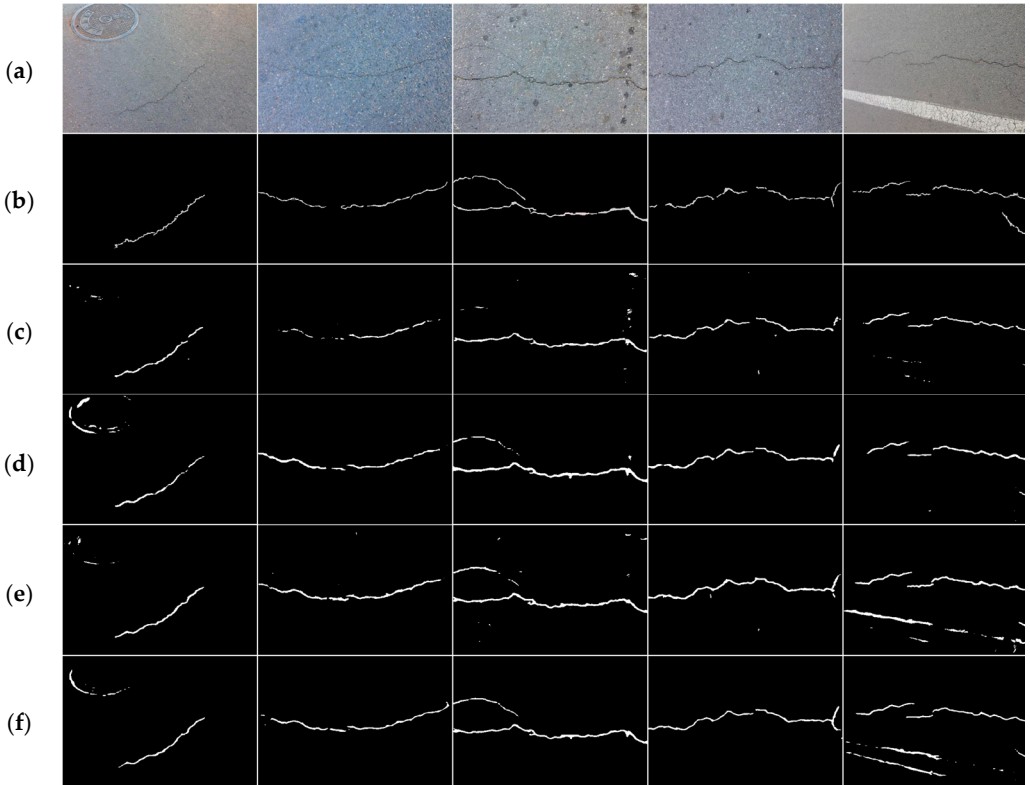

**Figure 3.** Results of pavement segmentation dataset. (**a**): input color images. (**b**): ground truth. (**c**): results of the UNet. (**d**): results of the UperNet. (**e**): results of the ResUNet. (**f**): results of the Pointrend.

### 3.3. Ablation Study

In this section, we performed ablation experiments by the proposed method on the pavement crack dataset to verify the effectiveness of the proposed architecture, i.e., the attention mechanism of the encoding layer (Att-UNet), the addition of weight standardization in the Att-UNet decoding layer (AttWS-UNet), the addition of multiscale Laplace

residuals in the AttWS-UNet decoding layer (AL-UNet), and add the PAN auxiliary head module in AL-UNet (ALP-UNet). We trained them with the same crop size, batch size, and the same loss function. We used 150 test images to test the segmentation results. The results of the qualitative comparison between these methods are shown in Tables 1 and 2 and Figures 3 and 4. It can be seen that the previous methods are susceptible to the effects of cracked unexpected interfering objects, while the ALP-UNet with CBAM module, Laplacian pyramid, and PAN structure reduces such effects and achieves better performance. Therefore, we believe that ALP-UNet can effectively reduce interference and can better extract detailed features.

**Table 2.** Comparison results of ablation experiments on different modules.

| Architectures | Recall | Precision | mFscore | mPA | mIou | Dice |
|---|---|---|---|---|---|---|
| Pointrend | 0.7559 | 0.5717 | 0.8234 | 0.7846 | 0.7372 | 0.6510 |
| Att-UNet | 0.7987 | 0.5491 | 0.8232 | 0.7735 | 0.7368 | 0.6508 |
| AttWS-UNet | 0.7965 | 0.5514 | 0.8237 | 0.7747 | 0.7373 | 0.6482 |
| AL-UNet | 0.8125 | 0.5653 | 0.8313 | 0.7817 | 0.7459 | 0.6667 |
| ALP-UNet | **0.8208** | **0.5802** | **0.8379** | **0.7889** | **0.7464** | **0.6798** |

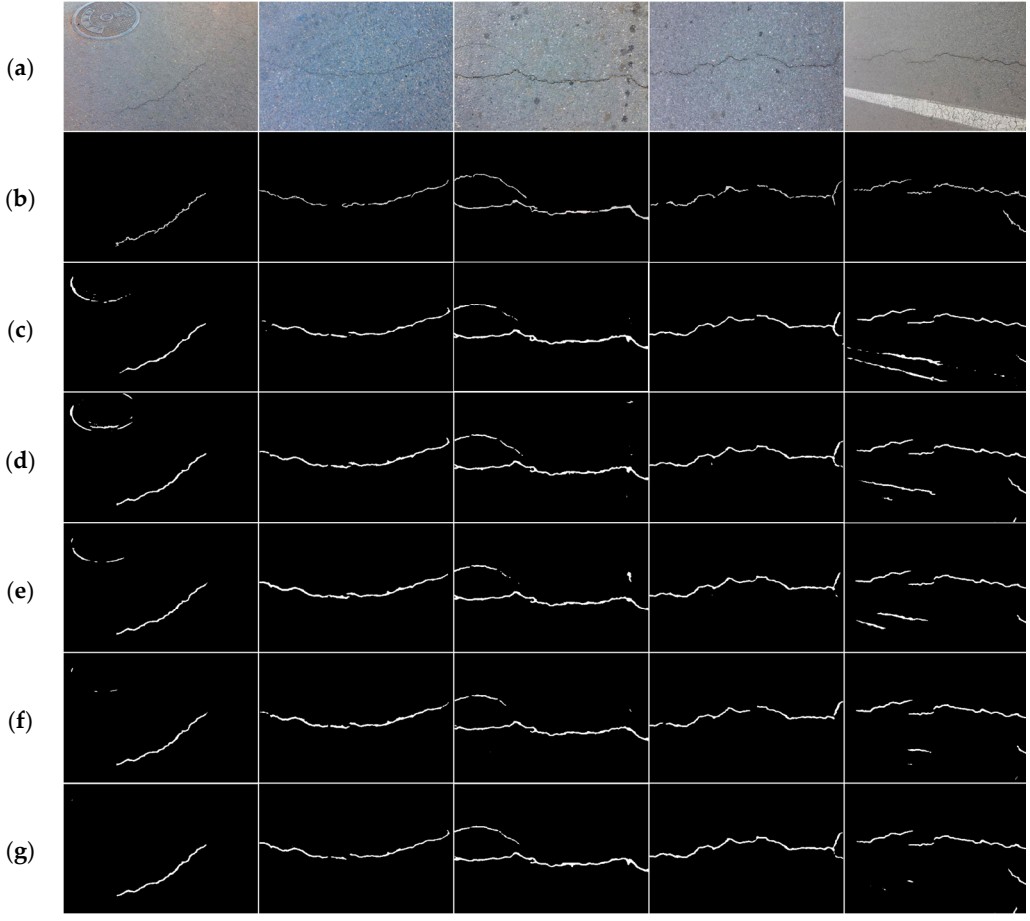

**Figure 4.** Results of the pavement segmentation dataset with different modules added. (**a**): input color images; (**b**): ground truth; (**c**): results of the Pointrend; (**d**): results of the Att-UNet; (**e**): results of the AttWS-UNet; (**f**): results of the AL-UNet; (**g**): results of the ALP-UNet.

From the above evaluation results, it can be seen that the segmentation results of UNet, UperNet, ResUNet, and Pointrend were all affected by manhole covers and stains on the pavement, and the Pointrend algorithm, which performs best in the evaluation metrics in Figure 3, was also severely disturbed by manhole covers and lane lines. The

output accuracy of the model can be improved by adding a CBAM attention mechanism in each coding layer stage. By adding weight standardization and multilayer Laplace residuals in the coding layer, it can be seen from the segmentation results that not only the segmentation accuracy of pavement cracks is improved but also the interference of other factors is reduced. By adding the PAN auxiliary head, our model achieves the best results in mPA and mIou metrics, and the error between the predicted and actual values of the proposed model in this paper is minimized, effectively reducing the interference of other factors on the pavement on the segmentation accuracy, and provides the best performance of the mentioned model in each metric. The results validate the effectiveness and superiority of the pavement crack segmentation model based on attention mechanism, weight standardization, Laplace pyramid, and PAN-assisted head.

## 4. Conclusions

In this paper, we proposed a novel pavement crack segmentation network based on deep learning. The main idea of this method is to improve segmentation accuracy and reduce the influence of pavement distractors on segmentation results by enhancing the feature extraction and fusion capabilities of the UNet network. Specifically, we added a CBAM attention module to capture the crack information more effectively during model training. We added weight normalization to the decoding process to stabilize the training process and improve accuracy. The boundary information in the feature map was fused into the decoding layer using multiscale Laplacian residuals to refine the segmentation results. We used the PAN structure to assist training by generating auxiliary supervision signals to further improve the segmentation accuracy. Finally, we chose a reasonable loss function for training, and the experimental results showed that our method achieves significant advantages in several metrics compared to existing methods. Our method can effectively segment small cracks and reduce the oversegmentation problems caused by lane lines or other objects. This paper provides a novel and effective technical method for the field of pavement crack image segmentation, and also provides insight and reference for other similar problems. In future work, we plan to apply our method to more complex scenarios with different types of pavement cracks and disturbances, and explore more effective network architectures and crack segmentation methods.

**Author Contributions:** Conceptualization, X.G.; methodology, Y.Z. and X.G.; software, X.G.; validation, X.G. and H.Z.; formal analysis, X.G.; investigation, X.G.; resources, Y.Z and H.Z.; data curation, Y.Z. and X.G; writing—original draft preparation, X.G.; writing—review and editing, X.G. and H.Z; visualization, X.G.; supervision, Y.Z.; project administration, Y.Z.; funding acquisition, Y.Z. All authors have read and agreed to the published version of the manuscript.

**Funding:** This research received no external funding.

**Data Availability Statement:** Not applicable.

**Conflicts of Interest:** The authors declare no conflict of interest.

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
