# Peer review of "Deep Learning-Based Semantic Segmentation Methods for Pavement Cracks"

_information, doi:10.3390/info14030182_

Round 1

Reviewer 1 Report (Previous Reviewer 1)

Please see the file attached.

Author Response

Dear reviewer, please see the attachment.

Reviewer 2 Report (New Reviewer)

In this paper, an interesting pavement crack segmentation network is proposed. The main idea of the method was to improve the segmentation accuracy and reduce the influence of pavement interferents on the segmentation results by adding a CBAM attention module to the coding process of the UNet network, adding weight normalization to the decoding  process to improve the accuracy, and using multi-scale Laplacian residuals to fuse the boundary information in the feature map to the decoding layer. Additionally, the PAN structure was used to assist training to further improve the segmentation accuracy. The experimental results from the above pavement crack dataset proved that the proposed method is effective for segmentation of pavement cracks. 

The work is well organized, though I would marge two first chapters into one. In my opinion both research description, methods and results are correctly presented, while conclusions summarize the work conducted. I would expect separate discussion section in which authors discuss the obtained results.

In section 2.1, I propose to add some older detection methods as well, proving that the authors have followed the literature carefully, one paper sticks in my mind (https://doi.org/10.1016/j.proeng.2017.02.004), which I recommend to the authors as one of the additional references.

Additionally, I recommend changing Figures 3 and 4 so that maybe Authors swap the rows with the columns, and then mark each column with the letters (a)-(g) at the bottom. Or at least rows lines (a)-(g) and later use these labels in the figure captions.

What do the authors intend to do in the next research step? What is the practical and industrial use of the output of their work.

What do the authors intend to do in the next stage of the research? What is the practical and industrial use of the achievements of their work.

My last, but the most important remark. I recommend to emphasize the novelty of the presented work in the last paragraph of the Introduction. A novelty in the context of proposed methods, but also in the scientific context. This is what I miss most about this job. What is the contribution of the authors to the development of science, except for the very efficient use of existing and generally known tools, such as deep learning networks, for specific applications, in this case, the analysis of the image of the pavement surface with damage in the form of cracks?

Round 2

Reviewer 2 Report (New Reviewer)

I believe the manuscript has been greatly improved.

I recommend accepting the article for publication in present form.

This manuscript is a resubmission of an earlier submission. The following is a list of the peer review reports and author responses from that submission.

Round 1

Reviewer 1 Report

The authors propose the deep learning-based segmentation model including the CBAM module, Laplacian residuals, and PAN structure. Particularly their backbone network is an AL-UNet. The proposed model can be powerful where edge information is dominant for the segmentation performance. The method was evaluated with in-house pavement crack dataset supporting the proposed method outperforms other methods. I have several comments to improve the model and the manuscript.

1. One of the reasons that you designed your backbone network, i.e., AL-UNet, is to effectively use the edge information. However, there are some studies that can more utilize edge information at U-Net. Modified U-Net ("Modified U-Net (mU-Net) with incorporation of object-dependent high level features for improved liver and liver-tumor segmentation in CT images." IEEE transactions on medical imaging 39.5 (2019): 1316-1325) is one of powerful method to grab the edge information of the original input image. So can you combine your method and mU-Net or compare the performance?

2. Please clarify the way to connection between your AL-UNet and PAN structure. It is not easy to understand which features go to where.

3. In your metrics, PA0 is the specificity, PA1 is the precision, and mPA is the averaged precision in the traditional terminology. So, with the same manner, please calculate the recall (sensitivity) and please change into the traditional terminology to avoid confusion.

4. For the Table 1, please show the number of parameters whether each method has similar numbers. Besides, the differences of the scores look very small. Please conduct the statistical analysis whether there is statistical meaning.

5. There are some types. Please remove the “-” in line 34, page 1. Furthermore, please do not use “I” in the scientific journal. It looks unofficial article. Please replace “I” with “we”.

Reviewer 2 Report

The article describes a new DNN architecture for the segmentation of pavement cracks.

The topic fits this journal but would be better to aim other journals with more specific scope such as deep learning, computer vision or even more application orientated journals as Applied Sciences or Engineering Applications of Artificial Intelligence.

The main ideas are to use and integrate known techniques for the detection of the small cracks on pavement images. Those techniques are: UNet, Laplacian pyramid, CBAM and PAN. The technical content, regarding these ideas, would be satisfactory and the paper could be accepted for publication but the presentation, the methodology details, the comparisons, the language usage are far from an acceptable journal article.

Thus, I propose the rejection of the paper.

Some of the main issues:

-          The paper targets only pavements. Other surfaces have similar cracks, why do they limit their approach to pavements? How other very similar approaches would perform for their purpose?

-          The classification in the review part is not very watchful, methods should be classified also based on the used sensory information such as (single lens cameras, stereo, structured light, TOF, etc.). Also, the class as “Digital image processing-based methods” is wrong since the other class also includes image processing aimed models. Classical, or hand-crafted feature based methods, or something more adequate should be used…

-          The literature review is very narrow, does not really analyze and criticize other papers. Their own idea should be finally compared to other approaches.

-          YoloV4 has PAN and Li [Li2022] added CBAM: how your method compares to them in performance, is there a difference how you applied these specific techniques?

-          There are lots of new papers with similar ideas even not mentioned in this article, f.e. [Zhang2020][Chen2023].

-          English usage is not satisfactory. It is not just the large number of typos, but the grammatical structures are of low quality. (e.g. “In the training phase, online data augmentation was performed, and to reduce the effect of overfitting problems.”, “König J proposed Architecture”, or “, the cracks on the highway will affect the traffic and collapse”, ….)

-          There is no information about the dataset. Even for a conference paper this is not acceptable, not even mentioning that there are several crack datasets [Kulkarni2022].

-          “We use 20 test images for evaluating the annotated 235 ground truth.” 20 images are very far to make conclusions when comparing methods. How were they selected, etc…

-          Mathematical notations are not precise. F. e. in Eq. 1. E_i is explained but Ei is used, bracket is needlessly used, i>0 and i=1 or i=2,3,4 on the other side has no sense at all.

-          Abbreviations should be written out at their first appearance… MLS, HOG, CBAM, SE,

-          Ablation study is not clear. ALP-UNet contains all proposed modifications to UNet – I guess. Then in an ablation study each modification should be cut individually and results are to be compared. It is not clear what is included in all other versions compared.

 Since the time to make this review is hardly limited, I think I stop my list here. It is enough to prove that the paper should be rejected in the present state and form. Maybe the ideas are good, the results could even overperform other methods, but it is not presented and proved the right way in this article.

Kulkarni, S., Singh, S., Balakrishnan, D., Sharma, S., Devunuri, S., & Korlapati, S. C. R. (2022). CrackSeg9k: A Collection and Benchmark for Crack Segmentation Datasets and Frameworks. arXiv preprint arXiv:2208.13054.

Zhang, Y., Huang, J., & Cai, F. (2020). On Bridge Surface Crack Detection Based on an Improved YOLO v3 Algorithm. IFAC-PapersOnLine, 53(2), 8205-8210.

Chen, L., Yao, H., Fu, J., & Ng, C. T. (2023). The classification and localization of crack using lightweight convolutional neural network with CBAM. Engineering Structures, 275, 115291.

Zhang, J., Qian, S., & Tan, C. (2022). Automated bridge surface crack detection and segmentation using computer vision-based deep learning model. Engineering Applications of Artificial Intelligence, 115, 105225.

Liu, H., Yang, C., Li, A., Huang, S., Feng, X., Ruan, Z., & Ge, Y. (2022). Deep Domain Adaptation for Pavement Crack Detection. IEEE Transactions on Intelligent Transportation Systems.

Li, L., Fang, B., & Zhu, J. (2022). Performance Analysis of the YOLOv4 Algorithm for Pavement Damage Image Detection with Different Embedding Positions of CBAM Modules. Applied Sciences, 12(19), 10180.